# Extracurricular Activities in Higher Education and the Promotion of Reflective Learning for Sustainability

**Ariane Díaz-Iso** **, Almudena Eizaguirre \* and Ana García-Olalla**

University of Deusto, Avda. de las Universidades, 24, 48007 Bilbao, Bizkaia, Spain
\* Correspondence: almudena.eizaguirre@deusto.es

**Abstract:** The objective of higher education institutions is to integrate reflective learning that contributes to the development of a greater awareness among individuals of the importance of facing the 21st century's sustainability challenges. This paper analyzes the impact of an extracurricular volunteer activity in Tangier, Morocco in the development of student reflection at a Spanish university. To this end, two objectives were proposed: (1) to explore the students' primary reflections of the experience, and (2) analyze the students' perceptions of the importance of participating in the experience in order to develop reflective learning. In the study, in-depth interviews were conducted with 23 students who participated in the volunteer activity. Data analysis was carried out using Iramuteq software to conduct a descending hierarchical classification (DHC), and MAXQDA software to conduct a constant comparison analysis. This research highlights the value of voluntary extracurricular activities in the development of reflections that guide change in the beliefs, attitudes, and daily behaviors of students that ultimately result in sustainability. Due to this, not only is it considered essential that students participate in social projects, but also that they undertake these projects with peers and instructors who can create environments of support and trust.

**Keywords:** extracurricular activities; volunteering; higher education; reflection; reflective learning; sustainability; sustainable development

---

## 1. Introduction

Public institutions, social organizations, schools, and families may all contribute to the development of citizens who are more aware of the importance of working toward sustainable development goals. Among these, higher education institutions play a fundamental role in educating competent professionals who can work to achieve sustainability [1,2]. Therefore, it is necessary to promote educational practices that help students become aware of the importance of exercising active and responsible citizenship that responds to the sustainability challenges of the 21st century [3]. It may be that generating a paradigm shift that addresses the students' sustainability needs, aspirations, and concerns becomes essential [4].

In this process, it is important to promote both active participation and autonomous and self-regulated learning in order for students to develop their reflective capacities. Reflection enables students to become aware of their strengths and weaknesses, provide solutions to complex situations while avoiding working by trial and error, face situations of uncertainty, to reformulate knowledge, practice and, most importantly, develop critical thinking skills and transform life experiences into learning [5–8].

In the field of curriculum development, researchers have shown an increased interest in analyzing the concept of reflection in higher education [9] and in demonstrating the importance of reflective practice within the curriculum in order to develop attitudes that contribute to sustainable development [10,11]. With regard to research into extracurricular activities, recent evidence suggests

the relevance of these activities in the improvement and development of reflective skills [12–14]. However, in-depth studies exploring students' experiences of self-reflection related to extracurricular activities (ECA onwards) are much needed [14].

In recent years, universities have aimed to integrate sustainability-related curricular and extracurricular reflective learning. Related to this, this research examines the emerging role of an extracurricular volunteer activity in Tangier, Morocco for the development of reflective skills. Specifically, the objectives of this study are:

1. To explore students' primary reflections from their voluntary extracurricular experience.
2. To analyze students' perception of the importance of participating in a voluntary extracurricular experience in order to develop reflective learning.

## 2. ECA (Extracurricular activities) to Promote Reflective Learning

Higher education is currently framed within a new conception of education geared toward sustainability. This requires the use of teaching and learning methods that motivate and make students aware of the importance of sustainable development. Such education needs to include key issues such as sustainable consumption, poverty reduction, and disaster risk [2]. It is essential to design courses that not only focus on student learning, but also contain reflective learning content that invites students to reflect on their daily learning and take action from a responsible, holistic, and forward-looking perspective [15,16].

In 1984, Kolb [8] conceptualized learning as a process of creating knowledge through the transformation of experience using the experiential learning model. Kolb proposed a cyclical learning model that consisted of four stages: concrete experience, reflective observation, abstract conceptualization, and active experimentation. According to the model, a cyclical process has to be completed, where experience constitutes the basis of learning and reflection. According to Kolb, learning occurs when reflection allows the individual to meaningfully learn from their own experience [17].

From this perspective, learning is less a process of knowledge acquisition, and more a process of knowledge construction in which reflective learning is paramount. This reflective practice is a dialog between thinking and doing [18–20]. Students who integrate theory and practice develop certain skills that enable better understanding of situations and the creation of independent meaning.

The term reflection has been defined in many ways. Moon [21] described it as a mental process applied in relatively complicated or unstructured situations where no obvious solution exists. Ryan and Ryan [22] argued that reflection allows students to examine what they believe and who they believe they are. Dewey [23] pointed out that reflection originates in a state of doubt or perplexity and, therefore, is an act of search and inquiry to find material to remove doubt and get rid of perplexity. For this reason, we defend that the reflective process usually begins in destabilizing and confusing situations. In these situations, instead of acting according to trial and error, a situation is interpreted and understood through questioning and research. In addition, reflection allows for the possibility of being aware of one's own actions, and learning from and improving those actions [24].

In relation to the moment of reflection, Schön [18] pointed out that there are two important temporal aspects: reflection-in-action (within experience) and reflection-on-action (after experience). The first refers to the reflection carried out as the action occurs. The second gives meaning to an experience after it has occurred. Such reflection can occur in two stages of the experiential learning model: in the reflective observation stage, when an experience is given meaning, and in abstract conceptualization, when concepts or hypotheses are generated [25].

Reflection, therefore, can be defined as engaging in a cognitive process in order to learn from experiences [6,16,23] and can be conducted through individual inquiry or in collaboration with others. Regarding individual reflective learning, instruments such as portfolios [20] or reflective journals [26] are used, where levels of reflection can be evaluated through forms of narrative. In these instruments, the quality of reflection varies according to the learner's ability to ask the relevant questions that will lead to learning [27].

Regarding reflective learning in collaboration with others, this can involve either interaction with peers or with a specialized instructor who guides the process. Socio-cultural theory conceives learning as a process of joint construction that occurs in the course of interaction [28]. In this regard, dialog is an essential moment of encounter, allowing for the construction of shared meanings from experience [29]. Among other methods, it can be carried out through seminars [30], online forums [31], or focus groups [32].

In this type of reflective learning, communication, cooperation, and feedback between instructors and peers play a fundamental role. Students, when interacting with others, promote reflective processes that help them to better understand themselves, their needs and problems as well as their personal strengths and limitations. In addition, these processes enable students to identify sources and means of coping with personal conflicts, challenges, and uncertainties [6]. For this process to succeed, the creation of a climate, based on mutual trust and positive bonds, is essential in order to provide security to the participating students [27].

Reflection is a process that requires stimulation, reinforcement, supervision, and training [23]. Therefore, the role of the educational instructor is fundamental when generating reflective learning [10,33]. The instructor, through dialog and the creation of a climate of mutual trust, should create a learning environment where students feel comfortable in expressing their thoughts [34] and reflecting on their actions [18]. The instructor is responsible for introducing, developing, and nurturing reflective learning [33]. Peer interactions also play an important role in the learning process. Sharing reflections, feelings, ideas, and experiences with others is a fundamental step in giving meaning to the learning process and student experience [17,35], and ultimately creating reflective processes where the points of view of others are assimilated, exchanged, and analyzed [36].

In the past two decades, a large body of scholarly literature has been published dealing with the importance of reflective learning in curricular disciplines related to education [24], health [26], social work [35], and business [34].

These studies have mostly been limited to the curricular field. However, there is increasing interest in the potential for ECA to promote reflection. ECA are defined as voluntary activities that take place outside the class schedule [37], which complement curricular training [38] and contribute to the students' personal [39], professional [40], and social [41] development. These activities are classified into sporting, cultural, solidarity, spiritual and artistic activities, and student clubs [41–43].

ECAs stand out for their ability to create spaces for the development of conflict resolution skills [44–46] and critical thinking and reflection on ethical values [12,14,47]. Indeed, Schripsema et al. [13] concluded that students who participated in ECA had better reflective skills than those who were not involved in such activities. In addition, [14] argued that participation in ECA facilitated reflection and allowed students to obtain the maximum out of that extracurricular experience.

## 3. Methodology

### 3.1. Context

This research explored a six-day extracurricular volunteer experience in Tangier, Morocco in January 2019 and June 2019 that involved 23 students from a Spanish university. The experience put these students in contact with people from a developing country. The gross domestic product (GDP) of the two countries should be highlighted as a gross indicator of their economic development level. In 2017, Spain's GDP was USD$1311 billion and Morocco's GDP was USD$109.1 billion.

During the six-day experience, students have the opportunity to become acquainted with the reality of immigration on the other side of the European border, participate in social projects, experience a culture and customs different to their own, and live as a group. The ultimate goal of this extracurricular experience is to promote student reflective learning regarding attitudes about sustainable development and explore the role that reflective learning plays in fostering sustainable development. In order to achieve this objective, the students are expected to engage in guided daily personal reflection.

Over the course of the experience, the students were introduced to three social projects in Tangier: (1) the Dar Al Baraka project of Casa Nazaret, a foster home that has arisen to respond to the needs of 10 people with special needs and without families and/or economic resources; (2) the Dar Tika project, a reception center that aims to provide care, training, and social insertion to underage girls at serious risk of social exclusion and/or lack of protection; and (3) the Padre Lerchundi project, a foster home for children that seeks to promote the integrated development of children aged 6 to 16 years old. In addition, as recommended by Elverson and Klawiter [48], at the end of each day, after participation in these projects, the students were encouraged to engage in guided reflection with their instructor.

*3.2. Sample*

In total, 23 university students (19 women and 4 men, aged between 16 and 24 years) took part in the experience. Students who participated in the experience in January 2019 (*n* = 13) are referred to as Group 1, and those who participated in June 2019 (*n* = 10) are referred to as Group 2 (see Figures 1 and 2). The percentage of participants who had previous experience in the extra-academic field was high. On the one hand, of the total number of participants in Group 1, 10 people had previously participated in ECA and three had not. On the other hand, when referring to Group 2, six people had previously participated and four had not.

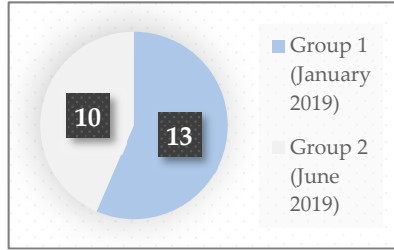

**Figure 1.** Characteristics of the sample. Distribution of the sample by group.

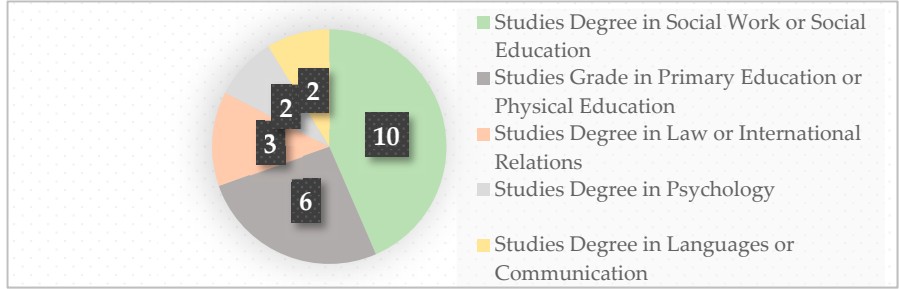

**Figure 2.** Characteristics of the sample. Distribution of the sample by degree studied.

*3.3. Research Tool: In-Depth Open Interviews*

The importance of the biographical-narrative approach in the social sciences is evident [49]. One of the options we considered for this study was to analyze the journals that the students were writing throughout their experience in Tangier. We believed that these journals were a valuable tool, as they reflected their day-to-day reflections. However, we were aware that many of the written reflections were personal and that the students may not want them to see the light. Given that the ultimate purpose of research into meaningful life experiences is not merely to understand the experiences people have had, but how their meaning is constructed [50], we decided to conduct in-depth open-ended interviews. As Chawla [51] defends, most recent studies of significant life experiences have been undertaken through structured or semi-structured interviews. When, thanks to our research, we encouraged participants to reflect on their own experience, we were at that moment contributing to a greater sense of authorship of the participants' lives, and the development and transformation of the participants was reinforced.

Therefore, in order to collect the necessary information, in-depth open interviews were conducted after the students had returned from Morocco. Interviewing is a research technique where an individual (interviewer) requests information from another individual or group of individuals (interviewee/s) by using a script of questions to obtain data regarding a specific matter [52]. The interviewer also encourages participants to go further by re-wording, re-ordering, or clarifying the questions. This technique was chosen for the present study as it has been proven to be adequate to explore the experiences of participants and the meanings they attribute to them [53].

The question script developed for this study was composed of 10 questions, as seen in Table 1. Questions 1 to 7 were developed based on Moussa-Inaty's [24] reflection guiding questions method. Moussa-Inaty's research showed the positive effect that certain questions can have in guiding student reflection (see Table 1). Additionally, consolidated criteria for reporting qualitative research (COREQ) were also considered when designing the question script [53]. Through these questions, the primary student reflections from the voluntary extracurricular experience in Tangier were collected.

**Table 1.** Reflection guiding questions used in the in-depth interviews.

| Reflection Guiding Questions |
| --- |
| 1.  How has this experience changed your way of thinking? |
| 2.  How could this experience change the way you act? |
| 3.  What surprised you the most about your experience? |
| 4.  What disappointed you the most about your experience? |
| 5.  If you had a chance to make a change (related to the experience), what would that change be? |
| 6.  What might some limitations be? |
| 7.  What do you plan to do further (related to the experience)? |
| 8.  How has participation in social projects promoted your reflection? |
| 9.  How has reflection with your peers enriched your own reflection? |
| 10.  How has having an instructor enriched your own reflection? |

Questions 8 to 10 sought to assess the students' perceptions of the importance of participating in the extracurricular experience as a way to develop their reflective skills. These questions were designed based on the findings of the literature review, which highlighted elements that were key to the investigation [6,33]. In addition, in order to guarantee the validity of the interview schedule, it was subjected to an assessment by three experts. Finally, participants were asked questions about their age, course, and grade.

### 3.4. Data Collection Procedure

Group 1 data collection was carried out between January 2019 and February 2019 and the Group 2 data collection was conducted in June 2019. First of all, the researchers contacted students enrolled in the experience and informed them about the purpose of the study, both verbally and in writing. As they all agreed on participating in the study, students were asked to sign a written informed consent. Subsequently, interviews were conducted in a climate of confidence and trust, allowing more personal and detailed access to the students' experiences. Given the unique nature of each participant's experience, in-depth interviews allowed each case to be investigated and for additional relevant questions to be asked in each individual context [54].

### 3.5. Data Analysis

Once the information was obtained, analysis of the information and effective treatment of the data were carried out. In order to provide a more complete picture [55], a data-analysis triangulation, consisting of the combination of two or more data analysis methods, was carried out [56]. Specifically, for Questions 1 to 7, Iramuteq software was used to conduct a descending hierarchical classification

(DHC) [57]. For Questions 8 to 10, MAXQDA software was used to conduct a constant comparison analysis [58]. Results obtained by each type of analysis were then combined at the interpretive level.

3.5.1. Instrument Used to Analyze Students' Primary Reflections on Their Voluntary Extracurricular Experience in Tangier (Questions 1–7)

In order to carry out a lexical analysis of the reflections expressed in the interviews, Iramuteq software was used to conduct the Reinert method. This analysis proposes that every discourse expresses a system of lexical words (or a group of words) that gives coherence and rationality to what the speaker expresses. The objective of the Iramuteq software algorithm is to perform textual data analysis, through the repetition of lexical footprints (word succession), in order to identify the most frequent lexical words shared by interviewees [59].

The software divides complete words such as verbs, nouns, adjectives, and adverbs from tool words such as articles, prepositions, pronouns, and conjunctions and includes only the former in the analysis. In the same way, it decomposes the corpus into elementary contextual units (ECU), which are one or two sentences long (30–50 words). In this way, the software performs an analysis of the complete words in each ECU. These words are then used to create a contingency table showing how the vocabulary is distributed by the ECU. From that table, the software creates a square matrix of distances and groups the ECUs according to the complete words they share [60].

Using the Reinert method, a DHC was performed to classify and group the ECUs into classes and consequently reveal the most characteristic vocabulary in each class. In this classification, each class was made up of different groups of words that were included based on the frequency of words already lemmatized, and the association with the class was determined by a chi-square value equal to or greater than 3. This provided a margin of error of <0.05 (of a degree of freedom = 1) [59]. Using this classification method, several classes were obtained based on words and ECUs with statistically significant values.

Operations within this program are transparent and replicable until the moment of interpretation when researchers assign a tag to each class depending on the most significant complete words and the ECUs. Likewise, the Reinert method, which uses independence tests, calculates the relationship between each class and independent variable. If one class has a significantly higher proportion of ECUs belonging to a variable, that class is considered to be associated with that independent variable [61]. In this case, the group was used as a variable; assigning Group 1 to students who went to Tangier in January 2019 and Group 2 to students who went in June 2019.

3.5.2. Instrument Used to Carry Out an Analysis of Perceptions Expressed in Relation to Voluntary Extracurricular Experience in Tangier (Questions 8–10)

MAXQDA software was used to carry out a constant comparison analysis of the students' perceptions. This type of analysis is commonly used when analyzing qualitative data [62]. This software is used in qualitative research for its ability to synthesize, sort, and organize information collected with selected instruments and to present research results [63]. The program organizes and examines information by means of categories formed by a system of codes.

This software was chosen because of the possibility it offered of creating three separate categories for analysis. For the purposes of this research, it was important to separate the students' perceptions regarding the value of the experience in Tangier; their perceptions of the role of their peers; and their perceptions of the role of the instructor in their reflective development.

A code system was then utilized to examine the data. With regard to the second research objective, the code system emerged from the literature findings. However, as the data were being coded, new (sub)codes were incorporated into the system and existing ones modified. Table 2 shows the final code system used to carry out the analysis.

**Table 2.** Code system used for the analysis of Questions 8 to 10.

| Categories | Codes |
|---|---|
| Importance of knowing the projects to develop reflection | To know; to think; to value; to be conscious; to compare; to change |
| Importance of living the experience with peers to develop reflection | To contribute; to enrich the reflection; to raise new situations |
| Importance of the instructor's role in developing reflection | Guiding; motivating; resolving doubts; climate |

## 4. Results

First, a DHC analysis of the reflections shown in the corpus (Questions 1 to 7) was performed using Iramuteq software (43,702 words; 24,031 words from the subcorpus of Group 1 and 19,671 words from the subcorpus of Group 2). Second, a mixed-method analysis of the perceptions expressed in Questions 8 to 10 was carried out using MAXQDA software. These questions formed a corpus of 11,822 words (5714 from Group 1 and 6108 from Group 2).

### 4.1. Descending Hierarchical Classification (DHC) Analysis of the Reflections Expressed in Questions 1 to 7

The first DHC divided the entire corpus into two different classes (or lexical worlds) (see Figure 3). Class 1 corresponded in a significant way ($X^2 = 34.81$, $p < 0.0001$) to the main reflections expressed by Group 1, and Class 2 to the reflections of Group 2 ($X^2 = 34.81$, $p < 0.0001$).

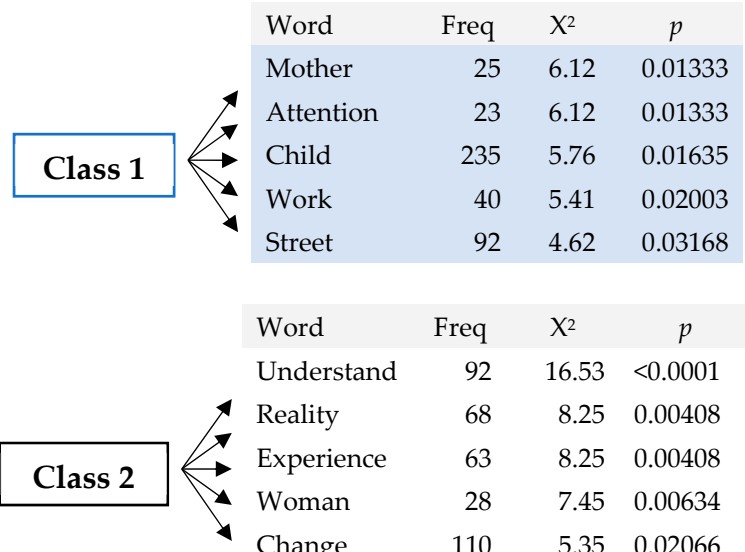

| Word | Freq | $X^2$ | $p$ |
|---|---|---|---|
| Mother | 25 | 6.12 | 0.01333 |
| Attention | 23 | 6.12 | 0.01333 |
| Child | 235 | 5.76 | 0.01635 |
| Work | 40 | 5.41 | 0.02003 |
| Street | 92 | 4.62 | 0.03168 |

| Word | Freq | $X^2$ | $p$ |
|---|---|---|---|
| Understand | 92 | 16.53 | <0.0001 |
| Reality | 68 | 8.25 | 0.00408 |
| Experience | 63 | 8.25 | 0.00408 |
| Woman | 28 | 7.45 | 0.00634 |
| Change | 110 | 5.35 | 0.02066 |

**Figure 3.** Distribution of classes and their respective units of meaning (DHC 1).

In order to more specifically identify the main reflections conducted in each class, a second DHC analysis of each group was carried out. The second DHC showed six classes for Group 1, and four for Group 2. The latter completely coincided with those found in the first group. Figure 4 shows the quantification of the classes and the statistical evaluation of the words in each group, based on their frequency, chi-square, and significance.

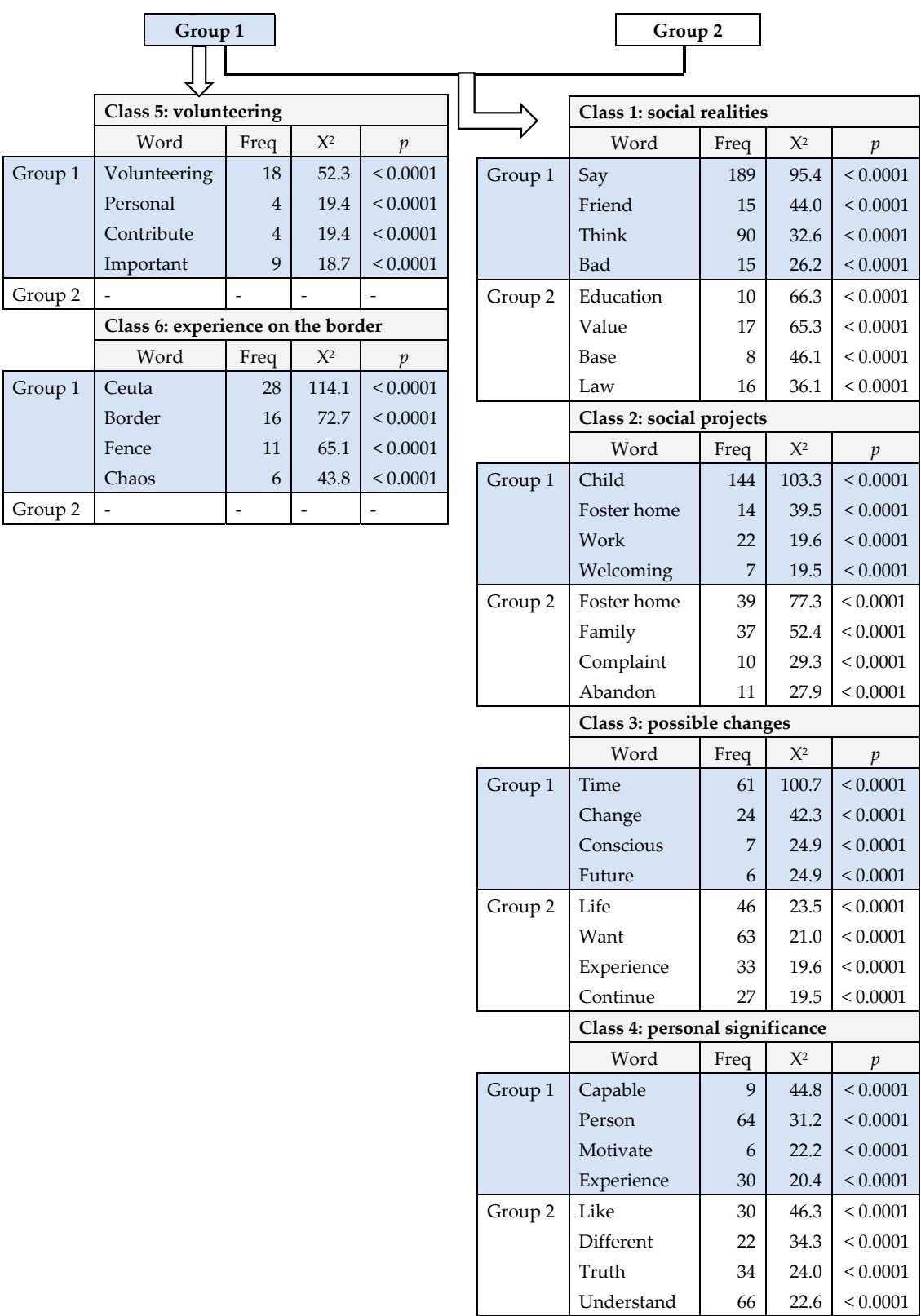

**Figure 4.** Distribution of classes and their respective units of meaning (DHC 2).

Next, the six classes (or lexical worlds) were broken down to gain an understanding of the different reflections shown by the individuals interviewed in each case. It should be noted that the analysis only highlighted those words with a significance level of $p < 0.0001$.

### 4.1.1. Class 1: Reflections on Different Social Realities

In this class (which consisted of 18.1% of the corpus), Group 1 reflected on its own privileges. In addition, Group 1 participants mentioned what they thought ($X^2 = 32.61$) people of very different realities were thinking: "Their life has to be very hard so that they have to think about all these things. Surely, they have lived experiences that are not even close to those I have lived at home or with my friends." (Participant 10, Group 1).

In Group 2, this class (which consisted of 13.3% of the corpus) placed value ($X^2 = 34.36$) on the right ($X^2 = 36.13$) to have health care ($X^2 = 32.95$) and public and accessible education ($X^2 = 66.37$). Likewise, this group reflected on the importance of achieving human rights in other countries so that social realities could be improved: "It would be interesting to create a residential center so that people do not live on the street and that guarantees them food and drink, as well as an education so that later they can have a dignified job that helps them to become self-sufficient." (Participant 21, Group 2). On the other hand, Participant 16 in Group 2 pointed out "I've become more aware of all reality, of everything. Because before I knew that there was poverty, that there were people on the streets sleeping... but until you see it in real life, you are not aware of the truth, of everything."

### 4.1.2. Class 2: Reflections on the Work Carried Out in the Projects

The second class, with 23.6% of the corpus in Group 1 and 23.9% in Group 2, was oriented toward reflections related to social projects.

Regarding the Dar Al Baraka project of Casa Nazaret for people with special needs, the reflections of Group 1 were oriented toward the work ($X^2 = 19.62$) carried out with people with special needs ($X^2 = 18.15$). Those of Group 2, however, were oriented toward the family abandonment ($X^2 = 27.91$) suffered by these people ($X^2 = 52.47$). Thus, one participant emphasized: "It has been a month since they went out in the streets! It is incredible. I wouldn't even be able to stay in for three days in a row . . . ." (Participant 19, Group 2).

With regard to the work carried out in the Dar Tika project with girls in need of help and protection, both groups reflected on the importance of becoming aware of one's own privileges as well as of valuing and making better use of one's personal situation. "One night a girl from the foster home took us up to the roof ( . . . ) and said to me, 'Do you know what that is?' and I said 'no'. 'Spain . . . I'm going there.' she said. I realized that people would give their lives to come to Spain . . . " (Participant 6, Group 1).

Finally, in relation to the work carried out in the Padre Lerchundi project with children at risk of social exclusion, both groups highlighted positive and hopeful reflections on the participants: "Even with all the terrible things they go through, they can still provide you with food, make your day happier, smile, share and make you feel at home." (Participant 15, Group 2) and "I would extend my stay longer and I would like to spend more time with those children and get to know them better." (Participant 23, Group 2).

### 4.1.3. Class 3: Reflections on Possible Organizational, Personal, and Social Changes

With regard to class 3, in Group 1, with 13.6% of the corpus, reflections were linked to possible changes at the organizational level. Above all, in both groups (with 31.9% of the corpus), students highlighted time as the main limiting factor of the experience: "We got to know many projects, but we didn't have the time to get to know the place and the people who were participating in each project. If we would have had more time, I think it would have been better, more enriching." (Participant 3, Group 1).

Group 1 also became aware ($X^2 = 24.98$) of the importance of making changes ($X^2 = 42.36$) at a personal level to bring about social improvement ($X^2 = 15.91$): "Even if we don´t like it, we are very ( . . . ) and I don´t consider myself being so, but whenever you travel there you see how everything you are supposed to do well, you actually don´t. You think we could all give a lot more." (Participant

13, Group 1), and make changes at the social level: "Teach everyone to be more supportive, more empathetic, more friendly, not to be indifferent . . . I believe that the necessary change in the society of the planet is educational." (Participant 9, Group 1). Group 2 reflected on changes that participants would like ($X^2 = 21.04$) to carry out on a personal level in their daily lives ($X^2 = 23.5$): "I am a person who goes very fast in my life and I am always thinking about the future and I don't think anything about the present, so I have to focus more on the present and live quietly day-to-day, not so disturbed nor looking at what I am going to do later." (Participant 14, Group 2), and on how to carry on ($X^2 = 19.56$) learning in other solidarity organizations.

### 4.1.4. Class 4: Reflections on the Personal Meaning of the Experience

Group 1 linked the fourth class (with 15.3% of the corpus) to personal reflections ($X^2 = 31.2$), encompassing how motivating ($X^2 = 22.28$) it had been to work with peers and other volunteers: "I used to say, 'What am I going to be able to do?', and there are people who are alone or with their partner and are doing everything they can, and it has made me think that 'I can also do something.'" (Participant 3, Group 1). Finally, Group 1 reflected on what they now needed in order to continue giving meaning to the experience: "Now I need to know what volunteer projects are here. I have seen what is being done in Tangier and Morocco, but what can I do from here?" (Participant 11, Group 1).

In the fourth class, Group 2 participants (with 30.9% of the corpus) reflected both on what the volunteering experience had meant on a personal level, and on how getting to know ($X^2 = 22.69$) different social groups had opened up new ($X^2 = 34.3$) professional perspectives: "Working with disabled people did not call out to me, it did not attract me much, it was seen as difficult and it is not an area that attracts me unlike immigrant children. Doing this experience has made me see that it is also an area that I like and in which I can participate." (Participant 17, Group 2). In any case, what is clear is that the experience was relevant to the students: "I believe that my way of thinking has changed... before I was still more likely to judge people without more, without even knowing them . . . ." (Participant 23, Group 2). In addition, Group 2 participants highlighted personal limitations to continuing to give meaning to the experience: "I would like to be able to do something in September, but next year is my last year and between the final career project, afternoon classes, and practices, I do not see it and as much as I see the first semester, but of course, that depends on the theoretical load that have the subjects . . . ." Participant 21, Group 2).

### 4.1.5. Class 5: Reflections on Volunteer Work Carried Out and Future Goals

The fifth class, with 17.1% of the corpus of Group 1, was linked to reflections on what it meant to participate in volunteer work ($X^2 = 52.37$): "It is as if you open your eyes that you do not have to go far to have to help or to take part in such projects, but that nearby there are also people who need help, you can lend a hand." (Participant 10, Group 1). In addition, Group 1 also reflected on how they could continue to have experiences that allowed them to contribute ($X^2 = 19.47$) on a personal level ($X^2 = 19.47$): "We are already thinking of going to Aitor (Aitor is the coordinator of the area of Solidarity and Cooperation at the University of Deusto) so that he can tell us what volunteer projects there are here." (Participant 6, Group 1).

### 4.1.6. Class 6: Reflections on the Lived Experience on the Border of Ceuta

In the sixth class, with 12.2% of the corpus, the participants reflected on the situation of chaos ($X^2 = 43.83$) that they experienced on the border ($X^2 = 72.7$) of Ceuta ($X^2 = 114.11$): "There is a difference between me coming here just for a day and these people living in a mountain during winter time to get through the fence." (Participant 2, Group 1) and "You realize that children are the future and they are living in very bad circumstances... Then... I would guarantee them the things they need so that they could have a dignified standard of living and have a future. What life can all these children have with such a hard experience on the border, many of them without a family to protect them?" (Participant 21, Group 2).

*4.2. Mixed Methods Analysis of the Perceptions Expressed in Questions 8 to 10*

The first analysis was carried out to identify the frequency of coded segments in each category. An initial approximation of the presence of each category in the corpus was then possible (see Table 3).

**Table 3.** Frequency of coded segments in each category.

| Categories | Group 1 | | Group 2 | |
|---|---|---|---|---|
| | Frequency | % Percentage | Frequency | % Percentage |
| Importance of knowing the projects to develop reflection | 19 | 43.18 | 28 | 41.79 |
| Importance of living the experience with peers to develop reflection | 15 | 34.09 | 28 | 41.79 |
| Importance of the instructor's role in developing reflection | 10 | 22.72 | 11 | 16.41 |

In order to deepen each category, an analysis of the coded segments in each category and their frequency is presented below.

4.2.1. Importance of Knowing the Projects to Develop Reflection

In Group 1, 92.30% of participants stated that the volunteer experience had proven to be a very useful tool in the development of their reflective skills. In Group 2, all participants highlighted its usefulness (see Figure 5).

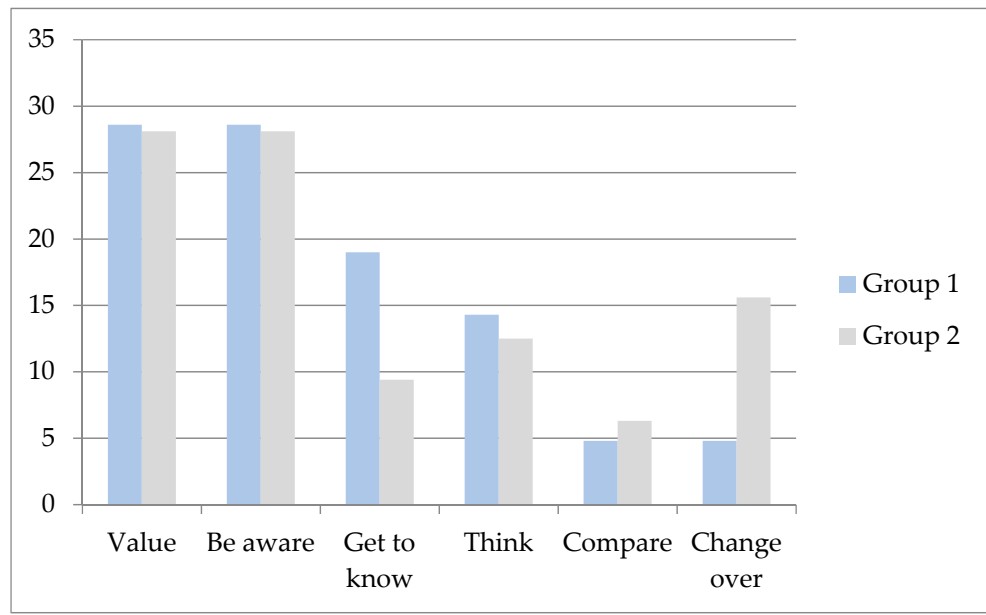

**Figure 5.** Frequency of codes in the category "Importance of knowing the projects to develop reflection.".

Participants cited that it helped to be aware (28.6% of respondents in Group 1 and 28.1% in Group 2) of different social realities and to value one's own privileges (28.6% in Group 1 and 28.1% in Group 2). Thus, as one participant emphasized: "The experience has taught me how to reflect on what I have and how lucky I am. It has also taught me how to value things more and now I see that you can just be happy with so little." (Participant 1, Group 1). "Visiting the projects so closely, meeting the people who work there dedicating their lives, has made me see my life in a different way..." (Participant 12, Group 1). In addition, some participants stated in this category that the experience had helped them to think about (14.3% in Group 1 and 12.5% in Group 2) and to know (19% in Group 1 and 9.4% in Group 2) other realities. In some cases, this enabled students to compare (4.8% in Group 1 and 6.3% in Group 2) the different realities of several countries and encouraged a change (4.8% in Group 1 and

15.6% in Group 2) of perspective: "It has helped me to get to know one of the most questioned religions in my country from within, being able to broaden the perspective I have of Morocco." (Participant 11, Group 1).

### 4.2.2. Importance of Living the Experience with Peers to Develop Reflection

Participants valued the experience as a very positive opportunity to reflect with their peers. In this category, participants focused on how important it was to create a space in which each person could contribute with different perspectives, thoughts, or ideologies (52.9% in Group 1 and 35.7% in Group 2): "My colleagues always showed me another point of view of the situations we lived there. I was able to get to know different ideologies of people of my age who see the world through different lenses." (Participant 11, Group 1) (see Figure 6).

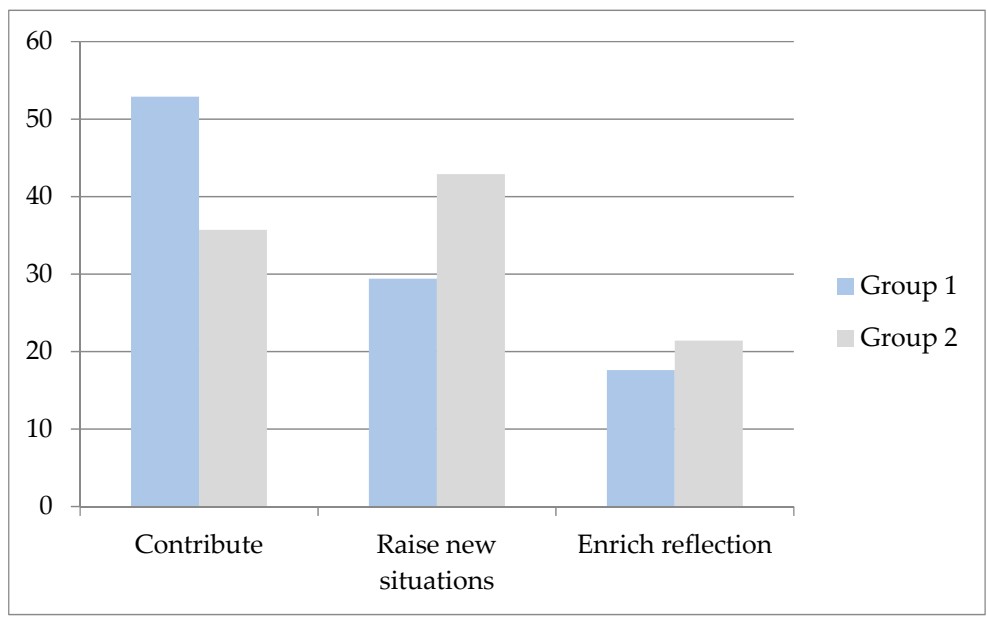

**Figure 6.** Frequency of codes in the category: "Importance of living the experience with peers to develop reflection.".

In addition, contributions from colleagues proved useful, on one hand, to raise new situations (29.4% in Group 1 and 42.9% in Group 2): "When exchanging opinions with them, I realized how significant many situations and ideas that I overlooked were, and vice versa." (Participant 10, Group 1). "It has been crucial for me to share this experience with my colleagues... Seeing them react, being able to contrast what we were each feeling... It has been a brutal experience, and my colleagues have played an important role in not feeling alone . . . " (Participant 5, Group 1). On the other hand, such contributions enriched reflection (17.6% in Group 1 and 21.4% in Group 2): "As we expressed our ideas and thoughts together, it also gave me the opportunity to speak, be heard and reflect on what others have experienced. This made the experience more enriching." (Participant 18, Group 2).

### 4.2.3. Importance of the Instructor's Role in Developing Reflection

The participants pointed out that the role of the instructor was very useful in guiding their reflections (42.9% in Group 1 and 45.5% in Group 2): "After dinner, we wrote down in a notebook what we have seen, felt, reflected on, lived, and observed during the day. This way, we had the opportunity to recall again what we had experienced during the day and to give it a second thought and to internalize it." (Participant 18, Group 2) (see Figure 7).

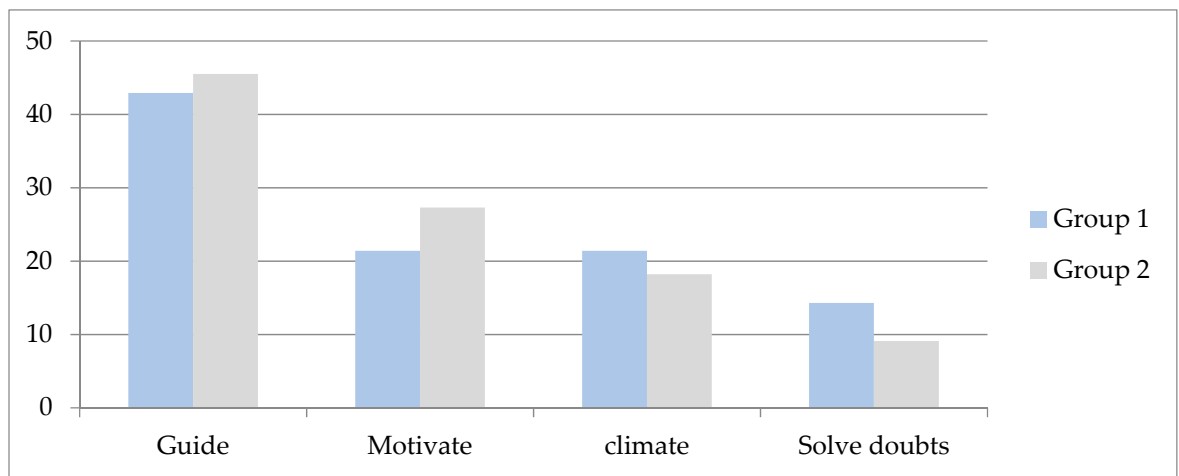

**Figure 7.** Frequency of codes in the category "Importance of the instructor's role in developing reflection.".

In addition, it was noted that the instructor had managed to motivate (21.4% in Group 1 and 27.3% in Group 2) the participants to reflect by creating a climate of trust and freedom (21.4% in Group 1 and 18.2% in Group 2): "I believe that Aitor has been a support and a guide, but I consider that the reflection was up to each one of us, how to carry it out, and how to interpret it. I consider that the role of the instructor has been based on an accompaniment on which we have been able to rely, if necessary." (Participant 5, Group 1).

Finally, there were those participants who believed that the instructor played an important role in solving doubts or clarifying ideas (14.3% in Group 1 and 9.1% in Group 2): "Aitor has made the whole experience easier for us on a daily basis. He encouraged us to make a reflection at the end of the day that helped us clarify our ideas." (Participant 7, Group 1).

## 5. Discussion and Conclusions

Returning to the objectives outlined at the beginning of this paper, the research involved a two-pronged approach:

(a) The global range of reflections experienced by Spanish students participating in a voluntary activity in Tangier, Morocco with people with special needs and minors at risk of social exclusion.
(b) The students' perceptions of the value of this experience in developing their reflective skills.

Taking into consideration the findings of the descending hierarchical analysis, regarding students' reflections on their participation in the project and their daily guided reflection, six distinct themes emerged. Four of these themes were found among the participants of both Group 1 and Group 2. These are: (1) reflections on different social realities; (2) the work carried out in social projects; (3) possible organizational, personal, and social changes that should be undertaken; and (4) the personal meaning of the experience. Two themes were found only among the participants of Group 1. These are: (1) the reflections on the voluntary work carried out; and (2) the lived experience on the border. The reason for this may be because the students of Group 1 visited the border between Morocco and Spain, while those in Group 2 did not. In their reflections, the students asked themselves why the social and economic reality in a place so close to Europe is so different. They wonder about the social, personal, and organizational changes that this type of societies requires. In their reflections, they showed how they had become truly aware of the lack of sustainable development on the ground. If sustainable development has most often been operationalized through a triangular vision of sustainability, which includes ecological, social, or socio-cultural and economic aspects, the socio-cultural and economic elements leave much to be desired, and the ecological element is far from being present. One of the participants pointed out "How is it possible for so much rubbish to be visible on the city streets?"

(Participant 1, Group 1), another pointed out "... the number of unschooled children on the roads." (Participant 13, Group 1), and another was impacted by "... the number of young people crowded on the border wishing to cross from Morocco to Spain in search of a new horizon of life." (Participant 9, Group 1).

The voluntary extracurricular activity developed in Tangier allowed students to reflect on themes that help educate citizens to be aware of and committed to the achievement of the Sustainable Development Goals [2]. What elements of the economic, socio-cultural, or ecological dimension come into play when witnessing so much inequality, poverty, and misuse of natural resources? To what extent are these societies compromising the well-being of future generations? Many of these ideas appeared in the reflection guided by the instructor who accompanied the group of students in their experience. This voluntary extracurricular activity gave both groups the opportunity to face situations of uncertainty, which made them aware of the different social realities faced by people with special needs and minors at risk of exclusion on the other side of the European border.

These results are consistent with other research [15], as this experience invites students to reflect on changes they can make to their own attitudes and actions from a perspective of responsibility, which contributes to an effective and well-founded grounding in sustainability. Participant 14 of the second group claimed, "I have realized the importance of a society that cares for the elderly and the sick, which requires a degree of social justice awareness that I have not been able to see here.". Other participants stated that the experience they had lived had made them change some beliefs "I have realized that schooling children is much more than preventing them from being idle in the streets and I wonder how much of the situation of this society is connected to the lack of education of children." (Participant 14, Group 2). On the other hand, in line with [22], students, placed in a destabilizing situation [21], were able to examine their self-image. In particular, they were able to examine their beliefs about who they believed they were and how they could undertake or continue to build a reality that increased hope and opportunity for people around the world.

With regard to the second objective, there was a high consensus that the development of reflection was one of the main objectives of the experience. In order to develop their reflective skills, the students highlighted three elements: (1) the importance of participating in social projects (with 42.48% presence in the corpus); (2) the importance of living the experience in a group (with 37.94% presence in the corpus); and (3) the importance of the role of the instructor (with 19.56% in the corpus).

As mentioned in the literature review, several authors [26,32] have indicated that the reflective process can be carried out individually or with external feedback. In this experience, the two groups pointed out the greater importance of reflecting with their peers than with the instructor. Thus, in line with [6], the volunteers stressed that interaction with their peers allowed them to take new approaches and enrich both their reflection and their experience. However, in line with Colomer et al. [10] and Peltier et al. [33], the participants also stressed the importance of the role of the instructor in generating a safe space of support and trust that invited reflection. Thus, in both groups, the fundamental task of the instructor was to guide the reflective process so that students could better understand the situation they were living through, and identify methods to face certain situations.

In order for this extracurricular activity to have greater impact, we consider it interesting to have a better balance between women and men participating in the experience as well as a greater variety of student profiles. We considered that the inclusion of engineering or business administration students would be of great value for the group. Apparently the proposal of this extracurricular activity has more demand from students of education, law, psychology or languages, but we considered that other profiles such as engineering or business would be enriched by the experience, and could bring other perspectives to the group. Another interesting element would be to reinforce the previous preparation to the experience (in depth study of the projects that are going to be visited, deepening in the socioeconomic reality of Tangier, etc.). Finally, the subsequent accompaniment to the experience would also reinforce the change of beliefs detected, and would help to ensure that it does not remain an isolated experience, and that it is part of the process of developing the competencies of university

students. The possibility of creating a learning community later on, with a monthly or bimonthly meeting, and continuing to collaborate with other types of extracurricular experiences in the country of residence also seems to us to be of interest. On the other hand, we would like to mention some obstacles that were encountered. First, due to the curricular load of the students, the experience was limited to one week and several students insisted on the appropriateness of lengthening the experience. Second, many of the participating students did not speak French, which prevented direct interaction with the people living in the centers visited. Third, and as we have already mentioned, the experience would have been richer with more varied student profiles (engineering or business students, etc.).

The primary conclusion of this research is that extracurricular activities that expose the student to real experiences of inequality and precariousness are an interesting element to contribute to deep and meaningful learning. In addition, the role of guided reflection in those experiences is very relevant, contributing to the integral human and professional formation of the participants.

In other words, the volunteer experience not only provides practical content that can contribute to the professional development of the individual [64], it also helps to develop values and attitudes that can guide personal development when carrying out sustainable development.

The study did not seek to generalize results, as quantitative studies do, but rather explore the impact that participation in ECA has on the development of reflective abilities. The research provides a detailed vision of the reflections extracted by students from a volunteering experience in Tangier, Morocco, and their perceptions of the importance of this experience to the development of their reflective skills. In addition, it adds new perspectives to an area that is increasingly the subject of investigation.

In the study presented, in-depth interviews were conducted with all of the participants after the experience. We believe that it would have been interesting to ask the same questions before having lived the experience to see how the answers changed before and after the extracurricular activity. On the other hand, we consider that it would be very interesting to interview the students one year after the experience, to see to what extent the impact detected is maintained over time. It would also improve the design of the research if we conducted in-depth interviews with the people in charge of managing the projects in Tangier to find out their perception of the value of this experience. Their opinion about what they see, hear, and observe in the students would be of great interest for the improvement of the extracurricular program.

Further research might explore the content of the individual reflective diaries written by students each night during the extracurricular experience. However, it is feared that informing students of the subsequent analysis of their diaries could generate bias in their reflections. A possible hypothesis to be contrasted is whether the fact of having previously participated in extracurricular activities has any impact on the participant's assessment of the new experience and of what nature. Likewise, in-depth interviews conducted both before and after the experience could provide interesting data, as would conducting identical interviews with students who have not participated in the experience, to compare their thoughts and perceptions. Finally, the development of a quantitative longitudinal study, based on a set of students who have the opportunity to live this experience each year, would prove very useful.

The findings imply that the supply of quality and structured ECAs [47] in higher education needs to be expanded. This need lies in the importance of ECA in fostering lifelong learning and in promoting reflection that enables students to enhance their skills to become better people and better professionals [65].

With regard to the future implications of this research, we would like to point out how structured and quality ECAs [47] in higher education can be an adequate path for the integral development of students. ECAs contribute to promoting a kind of reflection that helps students become aware of realities and situations that can make them better people and better professionals [65]. Therefore, public institutions must create laws with curriculum guidelines, university management teams, and faculty that encourage and support participation in ECA for the development of reflective skills, in order to produce citizens capable of facing the sustainability challenges of the 21st century.

**Author Contributions:** A.E. was the PI for the project and developed the paper's plan. A.E. and A.D.-I. conducted the literature review. A.G.-O. provided methodological support. A.D.-I. conducted the 23 in-depth interviews and A.E., A.D.-I., and A.G.-O. interpreted the outcomes of the in-depth interviews and wrote the paper.

**Funding:** This research received no external funding.

**Acknowledgments:** The authors gratefully acknowledge the Department of Solidarity and Extracurricular Activities of the University of Deusto for giving us the opportunity to analyze the impact of their activities.

**Conflicts of Interest:** The authors declare no conflict of interest.

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
