# Peer review of "Extracurricular Activities in Higher Education and the Promotion of Reflective Learning for Sustainability"

_sustainability, doi:10.3390/su11174521_

Round 1

Reviewer 1 Report

The theme is very interesting, the text is well written in general and the results are presented very well in a well-organized way. However, in the discussion of the results, the way the text is written give the idea that there is a gap between the results obtained and the assumptions that were intended to prove. I.e., the way the text is written gives the sensation that there is a lack of explanation between the 450-459 paragraph and the following 460-462 paragraph. Also, it would be important for the authors to better articulate the results they obtained with the following statements that make: “The voluntary extracurricular activity developed in Tangier allows students to reflect on themes that help educate citizens to be aware of and committed to the achievement of the Sustainable  Development Goals”, as well as with the expression:“as this experience invites students to reflect on changes they can make to their own attitudes and actions from a perspective of responsibility, which contributes to an effective and well-founded grounding in sustainability” (line 465-466).

-Another aspect is if it is true if the envolved students in this extracurricular activity were confronted with "a destabilizing situation" (line 468). I´m not sure.

-Is it not too strong to say that the results of this study come to bear to sure that: “The primary conclusion of this research is that deep and meaningful learning, carried out through guided reflection, makes a contribution to the integral human and professional formation of participants” (lines 487-489). Also, we can not infer so linearly the following issue: “With regard to the future implications of this research, the supply of quality and structured ECAs [47] in higher education needs to be expanded in order to encourage lifelong learning and promote reflection that enables students to enhance their skills to become better people and better professionals [56]” (lines 510-512).

If the authors can be more explicit I agree that the article will be a good text to be publicized.

Reviewer 2 Report

Research highlights the value of voluntary extracurricular activities in the development of reflections that guide change in the beliefs, attitudes,
and daily behaviors of a small sample of university students. Paper shows that educational practices that help students to become aware of the importance of exercising active and responsible citizenship that responds to the sustainability challenges. Any impact indicator of the program based on evidence of personal changes would be of interest to an international audience.

Beyond the hypothesis contrasts with chi square, it is recommended to provide a greater number of qualitative evidence. A greater emphasis on the quality arguments and life experiences of the students involved would bring more quality to the text. 

Extracurricular active participation, autonomous activities and self-regulated learning promote reflective capacities, provide solutions to complex 
situations and develop critical thinking skills in a interesting way to transform life experiences into learning. Please, characterize some obstacles encountered in the development of the experience and make some suggestions for improvement for the research design and the optimization of the quality of the extracurricular program.

It is recommended to include some reference to contemporary literature on life experiences and studies related to the field of higher education. A selection of biographical-narrative narratives showing examples of the acquisition of the aforementioned competences would be desirable to include in the paper.

Data analysis was carried out using software to conduct a descending hierarchical classification  and software to conduct a mixed methods analysis. A more detailed characterization of the techniques for collecting qualitative information and the methods of mixed methods used to analyze and triangulate information would be desirable to explain.

A more detailed description of the tasks and functions assumed by the students in the different projects would be desirable. A greater characterization of previous life experiences and the significant changes experienced in relation to these previous experiences.

It would be desirable to show some of the main limitations of the study in relation to the small sample used.
